# Early Waning of Maternal Measles Antibodies in Infants in Zhejiang Province, China: A Comparison of Two Cross-Sectional Serosurveys

**DOI:** 10.3390/ijerph16234680

**Published:** 2019-11-24

**Authors:** Ka Chun Chong, Yan Rui, Yan Liu, Tianyuan Zhou, Katherine Jia, Maggie Haitian Wang, Kirran N. Mohammad, Hanqing He

**Affiliations:** 1School of Public Health and Primary Care, The Chinese University of Hong Kong, Hong Kong, China; marc@cuhk.edu.hk (K.C.C.); yanyananan618@gmail.com (Y.L.); zhouty960821@126.com (T.Z.); katherine.jia19@imperial.ac.uk (K.J.); maggiew@cuhk.edu.hk (M.H.W.); 2Clinical Trials and Biostatistics Laboratory Shenzhen Research Institute, The Chinese University of Hong Kong, Shenzhen 518172, China; 3Zhejiang Provincial Center for Disease Control and Prevention, Hangzhou 310058, China; ryan@cdc.zj.cn

**Keywords:** measles, infants, China, MV, serosurvey, maternal, waning, vaccine

## Abstract

In China, children aged <8 months, who were expected to be protected by maternal antibodies before receiving the first dose of measles vaccine, were the age group with the greatest risk of infection in recent years. In this study, we evaluated whether infants yet to be age-eligible for measles vaccine had a sufficient seropositive level of maternal measles antibodies in 2009 and 2013. Blood samples were collected from infants aged <8 months through population-based serological surveys conducted in Zhejiang, China. Serum levels of immunoglobulin G measles antibodies were quantified using enzyme-linked immunosorbent assay. In 2013, the mean geometric mean titres (GMTs) of infants aged 4 to 8 months were below the seropositivity threshold (<200 mIU/mL), decreasing from 118.6 mIU/mL (95% confidence interval [CI] 83.0, 169.3 mIU/mL) at 4 months to 28.6 mIU/mL (95% CI 15.6, 52.3 mIU/mL) at 7 months. Antibody levels were significantly lower in 2013 than in 2009 starting from 5 months of age. In conclusion, infants aged 4 to 8 months are susceptible to measles due to low levels of maternal measles antibodies. It is thus suggested to provide infants with a supplementary dose on top of the routine schedule, and/or launch catch-up vaccination campaigns among young women.

## 1. Introduction

Measles is a highly contagious viral disease that can cause, encephalitis, diarrhea, severe pneumonia or even death. Even though a safe and cost-effective measles vaccine (MV) is available, measles is still associated with high morbidity and mortality, especially among children. According to World Health Organization (WHO) estimates, MV has prevented approximately 21 million deaths from 2000 to 2017. In 2017, approximately 110,000 people died from measles, the majority of whom were children aged <5 years [1].

In China, approximately 3 to 4 million cases of measles were reported per year before the introduction of MV in 1965 [2]. In 1978, following the WHO’s guidelines on Expanded Program on Immunization, MV was added to the routine immunization schedule and one dose of MV was provided free-of-charge to infants aged 8 months. Since 1986, a free second-dose of MV was offered to 7-year-old children, and in 2005, the timing of the second dose has been adjusted to 18 to 24 months after birth. With the intensified efforts to increase the two-dose vaccine coverage to around 95% in 2010, there was a major decrease in the incidence of measles from 200–1500 cases per 100,000 population per year before 1965, to approximately 6.8 per 100,000 population per year in 2000–2009. In 2009 and 2010, two national supplementary immunization activities (SIAs) were launched to provide MV to high-risk groups regardless their immunization history [3]. The incidence of measles further decreased to 2.8 and 0.8 per 100,000 population in 2010 and 2011 respectively.

Despite all the efforts to achieve universal MV coverage, China failed to achieve its goal of eliminating measles by 2012, most likely due to the change in epidemiology of measles infection [3,4,5,6]. One of the changes was that children aged <8 months, who were expected to be protected by maternal antibodies before receiving the first dose of MV (MV1), became the age group with the greatest risk of infection in recent years. The age-specific incidence of measles among young infants was higher compared to the other age groups [7,8]. Some local serological studies conducted among young infants found that early waning of maternal antibodies against measles might be the primary cause [9,10,11,12]. Similar phenomenon was observed in low-incidence and even measles-eliminated settings [13,14].

In China, where measles is endemic, the epidemiology of measles infection is changing over time and is creating obstacles to achieving the goal of measles elimination. In addition, only a few studies have monitored secular trends of maternal antibody levels among Chinese infants. In this study, we evaluated whether infants yet to be age-eligible for MV1 administration have sufficiently high maternal antibody levels to protect them against measles, and compared measles antibody levels among infants surveyed in 2009 and 2013.

## 2. Materials and Methods 

### 2.1. Setting

The study was conducted in Zhejiang Province in eastern China. Zhejiang has a population of approximately 54.8 million and a population density of approximately 537 per km^2^. It is a relatively affluent province with a per capita gross domestic product of approximately 9200 US dollars in 2012. It also has a well-established healthcare system.

In 1959, there were 770,000 cases of measles and >15,000 measles-related deaths reported in Zhejiang Province. In 1965, MV was introduced and was given to children aged <7 years. As a result, by 1977 the annual number of measles cases and deaths dropped to 100,000 and <300, respectively. In 2005, two-dose MV was introduced and was given to children at 8 months and 18 to 24 months of age. The reported annual incidence of measles decreased to a new minimum of 3.21 per 100,000 in 2006. There was a substantial increase in measles incidence in 2008. A province-wide SIA was carried-out in 2009, during which approximately 1.6 million middle-school, high-school and university students were vaccinated against measles. In 2010, the Zhejiang government adopted the recommendation from the 2006–2012 National Action Plan for Measles Elimination and conducted a province-wide SIA, targeting 2.5 million children between 8 months and 4 years of age. The reported MV coverage increased to about 97% in the target age group.

### 2.2. Data Collection

Samples were collected in 2009 and 2013 through population-based cross-sectional serological surveys and children were randomly selected from different cities. As this investigation was a part of the routine sero-epidemiology surveillance in Zhejiang province, ethics approval and informed consent were exempted by Zhejiang Provincial Center for Disease Control and Prevention (CDC). Children’s vaccination and infection histories were checked to confirm that they had neither been vaccinated against measles nor infected with measles. A minimum of 70 samples from infants aged <8 months were collected in each study year to ensure the precision of estimated overall seropositivity was within 10%, assuming that 80% of infants had a geometric mean titre (GMT) of measles antibodies >200 mIU/mL (the threshold for seropositivity) [8]. All data were kept confidential without patient identifiers.

The samples were sent to the WHO accredited National Measles Laboratory owned by Zhejiang Provincial Center for Disease Control and Prevention, where they were centrifuged and stored at −20°C until testing. Enzyme-linked immunosorbent assay (ELISA) (SERION ELISA classic measles IgG, Institut Virion/Serion GmbH, Wurzburg, Germany) was used to quantify serum levels of immunoglobulin G (IgG) antibodies against measles. The resulting optical density values were converted into titre units (mIU/mL) based on a calibration curve generated from standard serum. GMT of measles IgG antibody >200 mIU/mL, which is considered as a seropositive level by convention, is defined as seropositive in this study. Serum samples were prepared according to the manufacturer’s instructions.

### 2.3. Data Analysis

In order to assess whether infants maintained seropositive levels of measles antibodies up to the age of 8 months, a linear regression line was used to show the trend in the scatterplot of log-transformed GMT against age in days for each of the two years. The 95% confidence intervals (CI) of the regression line were calculated to assess whether the age-specific GMTs were significantly below the seropositivity threshold. To compare the 2009 and 2013 measles GMTs at different ages, picked-points analysis was used to assess the significance of the difference in mean GMT at 1, 3, 5, and 7 months of age [15,16]. *p*-values < 0.05 were considered to be statistically significant. All data analysis was done using SAS software, Version 9.4 (SAS Institute, Cary, NC, USA).

## 3. Results

A total of 77 and 78 samples were collected from infants aged <8 months in 2009 and 2013, respectively. The mean overall GMTs of measles antibodies were 981.5 mIU/mL (95% CI 673.7, 1289.0 mIU/mL) in 2009 and 454.7 mIU/mL (95% CI 274.0, 635.4 mIU/mL) in 2013. The overall seropositivity was 62.5% (95% CI 51.9%, 73.1%) in 2009 and 39.7% (95% CI 28.9%, 50.6%) in 2013.

Figure 1 and Figure 2 show the waning of measles antibody levels since birth until 210 days of life in 2009 and 2013, respectively. In 2009, the mean GMT decreased from 559.4 mIU/mL (95% CI 318.9, 981.4 mIU/mL) at one month of age, to 327.2 mIU/mL (95% CI 218.6, 489.9 mIU/mL) at 3 months of age, and further to 112.0 mIU/mL (95% CI 50.8, 246.6 mIU/mL) at 7 months of age (Figure 1). The proportion of infants who were seropositive decreased from 80.0% (95% CI 69.4%, 100%) at one month of age to 64.9% (95% CI 54.3%, 75.6%) at 7 months of age (Table 1).

In 2013, the mean GMTs remained >200 mIU/mL before 3 months of age. However, from the 4^th^ months onwards, the 95% CIs of the mean GMTs extended below the seropositivity threshold. The mean GMT decreased from 118.6 mIU/mL (95% CI 83.0, 169.3 mIU/mL) at 4 months of age to 28.6 mIU/mL (95% CI 15.6, 52.3 mIU/mL) at 7 months of age (Figure 2). The proportion of infants who were seropositive decreased from 80.0% (95% CI 55.2%, 100%) at one month of age, to 56.5% (95% CI 242.2%, 70.9%) at 5 months of age, and further to 39.7% (95% CI 28.9%, 50.6%) at 7 months of age (Table 1).

The waning of measles antibodies was more rapid and obvious in 2013 than in 2009. The mean change of GMT did not differ at one month of age, but a significantly lower GMT was observed in infants from the 2013 survey than in those from the 2009 survey when they hit 5 and 7 months of age (*p* = 0.03 and *p* = 0.007, respectively) (Table 2).

## 4. Discussion

Measles has yet to be eliminated in China and one of the major contributors might be the high measles incidence rate in children <8 months of age who are ineligible for MV1. In this study, we evaluated the sufficiency of measles antibodies in infants aged <8 months and compared the antibody levels between 2009 and 2013. We found that among infants aged 5 to 7 months, there was a significant decrease in measles antibody titres from 2009 to 2013; many children in this age range were seronegative, and therefore, at risk of measles infection before receiving MV1. These findings suggest that current control measures in China might be inadequate for elimination of measles. We thus recommend the officials to revise the vaccination schedule and provide an additional dose of MV to infants aged <8 months with low levels of maternal antibodies so as to close the immunity gap.

In Zhejiang province, infants aged <8 months had the highest measles incidence rates during 2013–2016, accounting for 20% to 35% of the total annual number of measles cases. In other provinces in China, such as Guangdong and Hubei, seronegative children were the main contributors to disease transmission during measles outbreaks [7,8]. Given the early waning of maternal measles antibodies, timely vaccination is critical for children to build up immunity against measles. Unfortunately, in China and elsewhere, a large proportion of children are not vaccinated against measles on time [17,18,19]. A study conducted in Tianjin Province found that 90% of infants did not receive MV1 on time [20]. Another survey conducted in eastern China found that >50% of children had delayed or missed their MV1 primarily due to difficulties in accessing healthcare services [21]. Other factors such as mother’s attitudes and knowledge, including vaccine hesitancy, were also associated with the timeliness of measles vaccination uptake. The impact of delayed MV1 administration can be substantial, and thus, it is particularly important to improve the delivery of vaccination services in rural areas where access to healthcare is limited. 

Several studies have mentioned the relationship between exposure to hospitals and measles incidence among infants in China. Ma et al. (2016) conducted a retrospective active case search and found a proportion of cases aged 0–7 months were infected in the hospitals in Kulun County of Inner Mongolia Autonomous Region [22]. A large case-control multisite study in China showed that the hospital visit was the most significant risk factor for measles infection in infants [23]. Another study from Gao et al. (2013) also found hospital exposure to be a significant factor contributing to measles infection among children aged less than one year [24]. Supported by our results and the nosocomial infections reported [23,25], we believe highly susceptible infants are one of the major causes leading to persistent endemic measles virus transmission in China. Infection control measures in hospitals shall be enhanced in order to reduce the risk of measles exposure to children, particularly among seronegative infants aged 5 to 7 months. For instance, infants and children with fevers and rashes shall be isolated promptly, preferably in negative-pressure rooms. Hospital staffs shall be given health education on infection control in order to improve their awareness of the risk of measles transmission among children and staffs.

Studies in other provinces in China have also shown that majority of infants were susceptible to measles infection in the months before they receive MV1. A serological study conducted in Guangzhou City within the Guangdong Province found that infants as young as 3 months of age were sometimes seronegative if their mothers had a low level of measles antibodies [9]. Even women who have acquired natural immunity or have been vaccinated might not be able to pass on sufficient maternal antibodies to protect their infants until 8 months of age. A study conducted in Tianjin Province from 2011 to 2015 showed that almost all infants lacked a seropositive titre during the period preceding MV1 administration, regardless of their mothers’ antibody titres [10]. In Jiangxi Province, <50% of children aged one year were seropositive [11], whereas in Qinghai Province, only 53% of infants aged 8 months were seropositive [12]. Nevertheless, these studies found that infants born to mothers with higher levels of measles antibodies tend to have seropositive levels of maternal antibodies for a longer period of time [9,10,12]. It is thus advisable to launch a catch-up measles vaccination campaign to provide women with an extra dose of vaccine as soon as they reach child-bearing age.

Even though two large-scale SIAs were carried out in China during 2009 and 2010, infants below the age of 8 months are still vulnerable to measles. Similar situation can be observed in low incidence countries in Europe where infants were prone to measles infection before receiving MV1 [13]. A study done in Belgium found that >50% of infants lost their measles antibodies within 4 months after birth [26]. Meanwhile, a review found that in sustained elimination settings, infants might be seronegative at birth if endemic transmission of measles virus has been interrupted for more than 12 months [14]. Since an increasing number of individuals are now acquiring their measles antibodies through vaccination instead of natural infection, the early waning of maternal measles antibodies in infants has become a new challenge for global measles eradication regardless of the setting.

Even though this study and several other studies have shown that maternal immunity to measles tends to wane as early as several months before MV1 uptake, whether or not to lower the age eligibility for the first dose of measles vaccine remains controversial. Some studies have shown that infants vaccinated at 6 to 9 months of age had lower seroconversion rates compared to those vaccinated at 12 to 15 months of age, not only because of maternal antibodies interference, but also because of their immature immune system [27]. Brinkman et al. [28] found that measles antibody levels in children who were given MV1 before 9 months of age were insufficient to provide long-term protection against measles beyond 4 years of age. Nevertheless, even though there is a risk of such “vaccine failures”, other studies have provided evidence that early vaccination against measles when the infants reach 4 to 8 months of age might provide survival benefit in rural settings [29,30,31]. In view of the lack of evidence for sustained immunogenicity if MV1 is given before 8 months of age, WHO has recently recommended that MV administration before 9 months of age should be considered as a supplementary dose on top of the two-dose routine MV schedule with the intention to cover the immunity gap arose from early waning of maternal measles antibodies [32]. This strategy might be useful in some settings where infants are at high risk of measles infection before MV1 uptake. Further experimental studies are needed to evaluate the efficacy of a supplementary dose of measles vaccine in infancy.

We compared the measles antibody levels in infants between the two surveys conducted 4 years apart. Results obtained suggest the idea of secular trends in maternal antibodies. However, a major limitation in our study is that we were unable to obtain information on maternal vaccination status nor history of maternal measles infection due to incomplete health records in the past in China, which makes it difficult to identify the reasons behind the waning effect of maternal protection against measles during the 4-year interval. However, given most of the mothers of the infants in our study were born between 1975 and 1995, this observation might be attributable to the increased MV coverage [33]. As vaccine coverage increases, mothers are less likely to be naturally infected with measles, and thus, have less antibodies available in the body for their child. Another major limitation is that we only evaluated the reduction in measles antibodies between the two serosurveys conducted in 2009 and 2013. Given that there were only two time points in our study, it is not possible to conclude that there is a secular trend of declining maternal antibodies in infants. Thus, we strongly recommend age-specific national serosurveillance for measles antibodies in China in order to monitor the population for susceptibility to measles.

## 5. Conclusions

In this study, a significantly reduced maternal protection was observed in infants aged 5 to 7 months during 2009 and 2013, majority of whom experienced waning of maternal antibody levels before they became eligible for MV1. We thus believe this immunity gap to be one of the reasons behind the failure of measles elimination programs in China. Adopting the WHO recommendation of providing infants with a supplementary dose of MV might be necessary in order to bridge the immunity gap and prevent resurgence of measles. Measures to reduce nosocomial infections (e.g., by providing isolation facilities) and timeliness of MV1 uptake (e.g., by improving access to vaccination in healthcare facilities) are important to reduce potential risk of local outbreak. If necessary, catch-up campaigns (i.e., providing woman with an extra dose of MV as they reach child-bearing age) shall also be considered in order to boost maternal antibodies, and thus, lengthen the duration of presence of maternal antibodies in their child.

## Figures and Tables

**Figure 1 ijerph-16-04680-f001:**
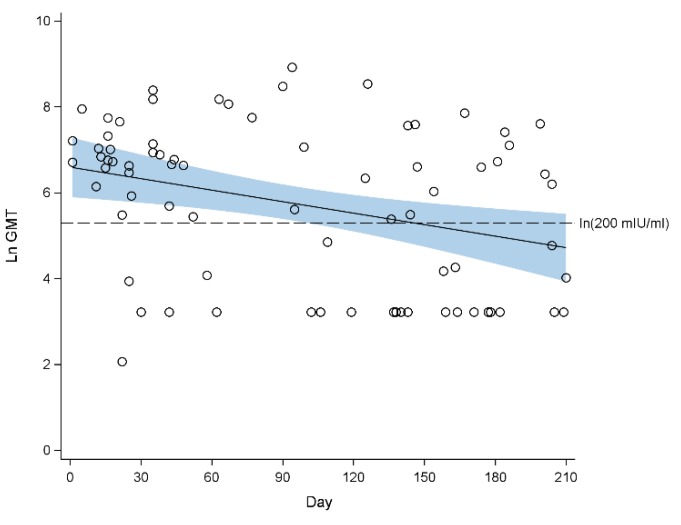
Scatterplot of natural log-transformed geometric mean titre (GMT) of measles antibodies against age in days from birth with fitted regression line for 2009. The blue band indicates the 95% confidence interval of the regression line. Half imputation was used for the values below lower quantification limit (50 mIU/mL). Seropositivity cut-off: ln (200 mIU/mL) = 5.30.

**Figure 2 ijerph-16-04680-f002:**
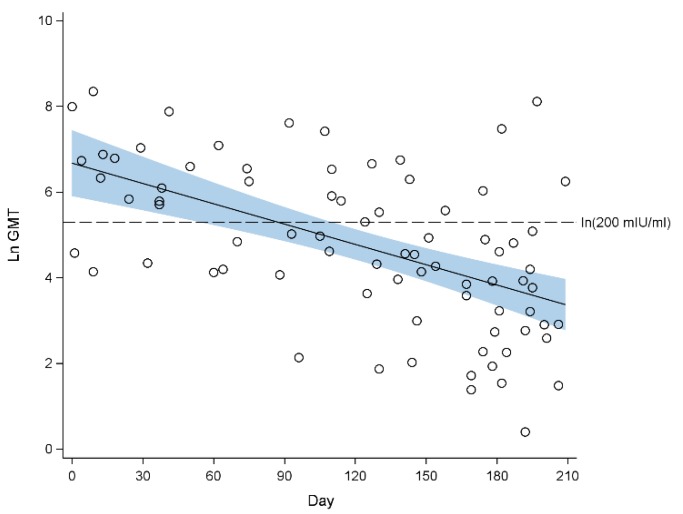
Scatterplot of natural log-transformed geometric mean titre (GMT) of measles antibodies against age in days from birth with fitted regression line for 2013. The blue band indicates the 95% confidence interval of the regression line. Seropositivity cut-off: ln (200 mIU/mL) = 5.30.

**Table 1 ijerph-16-04680-t001:** Estimated proportion (95% confidence interval) of seropositive infants by age in months for 2009 and 2013.

Month ^1^	2009	2013
1	80.0% (69.4% to 100%)	80.0% (55.2% to 100%)
3	83.8% (71.9% to 95.7%)	69.6% (50.8% to 88.4%)
5	73.2% (61.6% to 84.8%)	56.5% (42.2% to 70.9%)
7	64.9% (54.3% to 75.6%)	39.7% (28.9% to 50.6%)

^1^ The 1st, 3rd, 5th, and 7th months were assumed to be at day 30, 90, 150, and 210.

**Table 2 ijerph-16-04680-t002:** Mean difference (95% confidence interval) of geometric mean titre by age in months for 2009 to 2013.

Month ^1^	Mean Difference in GMT (mIU/mL) (95% CI)	*p*-Value
1	−70.1 (−348.3 to 583.5)	0.718
3	−136.8 (−218.7 to 8.8)	0.061
5	−117.1 (−151.5 to −51.8)	0.003
7	−82.8 (−100.5 to −35.6)	0.007

**^1^** The 1st, 3rd, 5th, and 7th months were assumed to be at day 30, 90, 150, and 210.

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
