# Peer review of "Early Waning of Maternal Measles Antibodies in Infants in Zhejiang Province, China: A Comparison of Two Cross-Sectional Serosurveys"

_ijerph, 2019, doi:10.3390/ijerph16234680_

Round 1
Reviewer 1 Report
Overall this is a really interesting study that contributes to a growing literature body that maternal antibodies to measles are declining rapidly.
Are you able to tell or did you ask if the child had a natural exposure to measles (to distinguish from the antibodies being derived from mother?)
You mention in discussion that infection control in hospitals is important (line 160). Have previous studies found a link between exposure to hospitals and measles disease among infants in China?
An explanation of why there is declining antibodies could be helpful - for instance, children nowadays are more likely to have mothers who were vaccinated but did not have an exposure to natural measles infection, which could reduce transfer of antibodies to children.
Did parents provide informed consent to be in the study? Or could this be exempted from certain ethical oversight because it falls under routine public health surveillance?
Minor comment - Figure 1 is a bit blurry. Would be best if you could natively export it as a figure with a resolution of at least 300 (or just plot it in Excel or another spreadsheet program). My preference if at all possible would be to keep the y axis in GMT (not log GMT) but to have the axis on the log-transformed scale (so that the distance between points is as you have it, but the numbers correspond to actual GMT values).
Reviewer 2 Report
The authors reported the decay of measles antibody in Chinese children <8 months of age in 2009 and 2013. Antibody levels were significantly lower in 2013 and seropositive rate decreased from 80% at one month, 56.3% at 5 months and 39.7% at 7 months. They suggested the supplemental measles vaccine dose at <9 months and among young women. I have several major comments.
They examined the IgG EIA antibody and they wrote in the Materials and Methods, Data Collection in page 3, lines 94-100. According to conversion, a GMT of measles IgG antibody >200 mIU/mL was defined as seropositive. Does this mIU/mL mean WHO reference serum PRNT titer? Protective antibody levels should be examined through PRNT because EIA does not always reflect the neutralizing activity. How to convert from EIA optical density to mIU/mL? In Figures 1 and 2, vertical axis represented Log GMT 0-10. The figure showed scatterplot of individual serum antibody titers, not GMT. Cut off level of >200 mIU/mL is lined between Log GMT 4 and 5. How to transform the titer? It should be explained in their legends. In the results, page 3, lines 112-113, overall GMTs were 981.5 and 454.7 mIU/mL in 2009 and 2013, respectively. As for in 2009, GMT at on month is 559.4 mIU/mL, 327.2 mIU/mL at 3 months, and 112.0 mIU/mL at 7 months. Why is the overall GMT higher than those at 1, 3, and 7 months? Similar to the overall GMT in 2013. In page 5, lines177-179, they wrote that measles vaccine can even be given to women during pregnancy in outbreak setting, if there is a risk・・・・・. These sentences should be deleted because a live vaccine is contraindication during pregnancy. In conclusion, they discussed the supplementary measles vaccine for young infants and young women. But I think it would be an imminent issue that >50% children had delayed or missed their MV1, as mentioned in lines 210-215. Before discussing the additional dose,
As minor comments:
Page 3, line96, -20oC should be changed to -20℃. Page 4, line 144, is Measures typographical error of Measles? Page 5, line 152, “a supplementary dose be” should be changed.
Round 2
Reviewer 2 Report
They responded most of my comments. But regarding Figures 2 and 3, I still have several comments. They wrote the vertical axis is Log GMT. Each circle represents the individual NT titer converted from the EIA values for the day from the birth. Not mean titer.
They mentioned that seropositivty cut-off level 200 mIU/mL is lined Log 5.3. I wonder why 200 mIU/mL is 10 5.3. 200 = 10 2.3
In Fig. 1, many samples were lined on the same level approximately 3.0 and only one below 3. But in Fig.3, they were scattered. Are their detection limits different in Fig. 2 and 3?
